# Impact of Higher Airspace Operations on Air Traffic in Europe

Oliver Pohling *, Lorenz Losensky , Sandro Lorenz  and Sven Kaltenhäuser 

Institute of Flight Guidance, German Aerospace Center (DLR), Lilienthalplatz 7, 38108 Braunschweig, Germany; lorenz.losensky@dlr.de (L.L.); sandro.lorenz@dlr.de (S.L.); sven.kaltenhaeuser@dlr.de (S.K.)

\* Correspondence: oliver.pohling@dlr.de; Tel.: +49-531-295-3892

**Abstract:** Historically, higher airspace has been used for military exercises and as transit for space vehicles. Riding on commercial space operations' coattails, more and more vehicles are under development that will make use of higher airspace resources. This will lead to increasing interactions with conventional air traffic since these new vehicles will have to transit through lower airspaces. The management of these operations is necessary to ensure the safe and practicable shared usage of these airspaces. This paper outlines an assessment of the impact of higher airspace operations on conventional air traffic in Europe. Initially, a synthesis of possible use cases was performed, and demand scenarios were developed that served as input to a fast-time simulation. The impact on air traffic was measured by means of flight efficiency parameters. The simulation results showed that the impact is dependent on the type of operation. High-altitude platform system flights and orbital launches cause the largest deviations in flight distance, flight duration and fuel consumption. Higher airspace operation parameters, including location, time, and duration, strongly affect the impact on the conventional air traffic.

**Keywords:** higher airspace operations; impact assessment; fast-time simulation

## 1. Introduction

In recent years, there has been growing interest in the commercial use of space. This will lead to an increasing number of space operations from a variety of spaceports and launch sites with many new vehicle types, such as super- and hypersonic aircraft, sub-orbital and trans-atmospheric vehicles, high-altitude platforms, and space vehicles [1]. These so-called new entrants transit through airspaces that are used by conventional air traffic along the way to their mission altitude. The intensification of operations by new entrants will lead to additional needs for airspace resources. To cope with these kind of operations, the Federal Aviation Administration has developed a concept of operations concerning the integration of commercial space into the National Airspace System of the USA [2].

In Europe, there is also a need to enable safe and efficient higher airspace operations (HAO). HAOs are defined as operations that take place at altitudes above conventional air traffic, which does not occur at altitudes above 50,000 ft, and that use the higher airspace as part of their mission or to reach space. The project 'European Concept of Higher Airspace Operations' (ECHO) addressed this need and delivered a concept of operations to integrate new entrants into the European airspace [1]. Within this project, a demand synthesis and impact analysis was conducted. The aim of this research was to estimate the demand of future HAOs that emerges from the various vehicle types of the new entrants and to assess the potential impact of the resulting HAOs on conventional air traffic. This paper addresses parts of the demand synthesis and impact analysis using a fast-time simulation tool where different use cases of HAOs are modeled. The focus of the impact assessment is on a regional scenario in Europe that already has declarations of intent and ambitions of operating higher airspace vehicles (HAV) within its airspaces. The impact is then assessed by a definition of flight efficiency parameters, and their changes are analyzed. The scope

of this study focuses only on nominal events, so failures or malfunctions of HAVs are not examined.

The paper is organized as follows. In Section 2, a brief overview is given about existing impact analyses regarding space operations or HAOs. The focus is on the usage of fast-time simulations to assess the impact. Section 3 outlines the synthesis of new entrants and derives possible use cases of HAOs, especially with relevance in Europe. The development of demand scenarios, which serve as a basis for the impact analysis, from these use cases is described in Section 4. The approach and methodology of the impact assessment is presented in Section 5. Additionally, the simulation setup is described. In Section 6, the results of the analysis are outlined. Afterwards, Section 7 features a discussion of the results. Finally, the conclusions are given in Section 8.

## 2. Existing Works

In the literature, there are several examples of assessing the impact of HAOs on conventional air traffic using fast-time simulations.

Young et al. [3] addressed the effects of prospective space operations on air traffic with a focus on the National Airspace System of the USA. They used the fast-time simulation software "Air Traffic Optimizer" (AirTOP) and identified changes to the flight efficiency parameters of the air traffic because of space operations in two forecasted years (2018 and 2025). The study comprised different launch locations in the USA, and the space operations covered types such as rocket launches, e.g., SpaceX Falcon 9, or sub-orbital flights, such as the Virgin Galactic SpaceShip 2. Another focus of the impact analysis was on procedural changes regarding the handling of space operations and separation from air traffic, as well as an analysis of effects to the sector throughput. The main findings were that the forecasted space operations lead to deviations regarding the air traffic's flight efficiency and violations in case of the manageable sector throughput. The usage of changed air traffic control (ATC) procedures can significantly decrease the impact of space operations.

Luchkova et al. [4] analyzed the traffic impact of a sub-orbital intercontinental space operation use case with a focus on the return of such a space vehicle (SpaceLiner) back to a European spaceport. The authors calculated the SpaceLiner's resulting hazard areas, which specify the size of an area where debris drop down in case of a non-nominal event, to measure the influence of space operations on air traffic. The hazard areas were modeled within the fast-time simulation tool AirTOP, and parameters such as the entry and exit count, flight duration, flight distance, and sector occupancy were observed to determine the interaction between hazard areas and air traffic. The traffic sample covered three days, each with a duration of 24 h. The results showed that the influence of such space operations can be quite substantial when following a conservative space traffic management approach. It has been suggested that the dynamic handling of hazard areas would be preferable to minimize the impact of space operations.

Kaltenhaeuser et al. [5] delivered an evaluation of the possibility of integrating air-launch operations in the European airspace. Based on information from an already-performed launch event in the USA, the resulting restricted airspaces concerning air traffic were transferred to an area over the North Sea in Northern Europe and afterwards were modeled in the fast-time simulation tool AirTOP. The study included only the restricted airspaces of an air-launch corridor that are within the European airspace, which included two drop-off zones for the first and second stage. The impact of this air-launch was measured by simulating one day of air traffic and activating the restricted airspace during times of low flight movements and during a peak hour. Affected aircraft had to reroute around the airspaces when necessary. The results showed that the integration of air launches is possible in Europe, but rerouted flights had increases in flight distance, delay, and fuel consumption in a low-single-digit percentile range.

Another analysis of the impact that space operations have on air traffic was given by Tinoco et al. [6]. Their paper focused on an air-launch use case where the space vehicle is carried by an aircraft to a specific altitude to perform the launch. The location of this

use case was set to be the Cecil Air and Space Port in Jacksonville, Florida (USA). The space operations of this use case were modeled as restricted airspaces in the fast-time simulation software Total Airspace and Airport Modeler by Jeppesen. The flight plan covered a duration of 24 h and was based on the day with the busiest interval of air traffic during the launch window to cover the worst-case impact. Various scenarios were investigated that modified the duration of the launch window and the number of restricted airspaces. Additionally, the flight plan featured two forecasted traffic samples to take account of changes in traffic volume. The analysis revealed that a reduced duration of airspace closures can limit the impact of air launches on air traffic. With reduced airspace closures, the flight delays, additional flight distances, and fuel costs decreased by about 60%. Furthermore, a change in the segregation management between space operations and air traffic resolved each conflict, but the conclusion only applies to this study as it is dependent on the location of restricted airspaces and the direction of air traffic flow.

Lehmann et al. [7] performed an analysis of the possibility to integrate space operations in the dense airspace of the United Arab Emirates (UAE). This study differed from the previously described ones in that an event model was used instead of a fast-time simulation tool to analyze the impact of space operations on air traffic. An air-launch operation, such as the Virgin Galactic SpaceShipTwo, was chosen as the use case since a declaration of intent exists. A peak hour of air traffic was examined to cover a worst-case scenario. Additionally, a potential spaceport location was identified, and hazard areas were designed to depict off-nominal events. After an initial assessment of the affected flights, a 15 min interval was analyzed during the peak hour. Lehmann et al. found that most of the affected flights departed from or arrived at airports in the UAE. Here, possible solutions are the delay of flights on ground or airborne holdings, whereas overflights need to be rerouted. The authors concluded that space operations can be incorporated in the UAE airspace.

This literature review has shown that the studies by Luchkova et al. [4], Kaltenhaeuser et al. [5], and Lehmann et al. [7] examined only a single use case of HAOs and their effect on conventional air traffic. They did not investigate different vehicle categories or multiple locations of HAOs. Young et al. [3] and Tinoco et al. [6] did account for that, but their operational sites were within the USA. Additionally, all of the impact studies listed here of HAOs on conventional air traffic focus on the impact of vehicles that are equipped with rocket propulsion. As a novel contribution, this paper will address various categories of HAVs, including high-altitude platforms, concomitant with multiple operational sites being located in Europe.

## 3. Synthesis of New Entrants

In this chapter, HAO-specific demand scenarios are developed. Our intelligence was gathered within the task "Demand Synthesis and Impact Analysis" in the project ECHO, which was funded by the European Union's (EU) Horizon 2020 research and innovation program.

As a result, the scenarios are later used as the reference for the impact assessment.

### 3.1. Use Cases

A comprehensive analysis of use cases covering all expected types of vehicles and operations was conducted, identifying the following main categories of vehicle operations with regards to HAO [8]:

- High-altitude platform system flights (HAPS);
- Orbital launchers (LAUN);
- A-to-A sub-orbital flights (ATOA);
- A-to-B sub-orbital flights (ATOB);
- From-orbit flights (FORB).

Based on the main HAO categories, the following sub-categories (and sub-sub categories) were designated [8] and are detailed in Table 1:

**Table 1.** HAO Use Cases.

| Use Case | Identifier | Sub-Category | Sub-Sub-Category | Example |
|---|---|---|---|---|
| HAPS | UC_HAPS_FB1 | Lighter than air (LTA) | Free balloon | Stratospheric balloon |
| | UC_HAPS_MB1 | LTA | Maneuvering balloon | Loon |
| | UC_HAPS_AS1 | LTA | Airship | Stratobus |
| | UC_HAPS_AC1 | Heavier than air | Fixed-wing aircraft | Zephyr |
| Launchers | UC_LAUN_SR1 | Direct launch | Sub-orbital expendable rocket | Sounding rocket |
| | UC_LAUN_DE1 | Direct launch | Expendable rocket | Ariane V |
| | UC_LAUN_DR1 | Direct launch | Semi-reusable rocket | Falcon 9 |
| | UC_LAUN_DR2 | Direct launch | Fully reusable rocket | Starship |
| | UC_LAUN_RP1 | Direct launch | Rocket plane | Skylon |
| | UC_LAUN_AE1 | Air launch | Expandable rocket | Launcher One |
| A-to-A | UC_ATOA_AL1 | Air Launch | Reusable air-launch rocket plane | SpaceShipTwo |
| | UC_ATOA_VR1 | Direct launch | Reusable rocket (Vertical takeoff and landing) (VTOL) | New Shepard |
| | UC_ATOA_RP1 | Direct launch | Reusable rocket (Horizontal takeoff and landing) (HTOL) | Lynx |
| A-to-B | UC_ATOB_SA1 | Supersonic aircraft | Air-breathing propulsion | Overture |
| | UC_ATOB_HA1 | Hypersonic aircraft | Air-breathing propulsion | Hermeus |
| | UC_ATOB_HS1 | Hypersonic spacecraft | Reusable rocket plane (Vertical Takeoff and horizontal landing) | SpaceLiner |
| | UC_ATOB_HS2 | Hypersonic spacecraft | Reusable rocket (VTOL) | Starship |
| | UC_ATOB_HS3 | Hypersonic spacecraft | Reusable rocket plane (HTOL) | MBB Sänger II |
| From orbit | UC_FORB_RV1 | Controlled | Re-entry vehicle | Dragon |
| | UC_FORB_SD1 | Controlled | Object de-orbit | Satellite |
| | UC_FORB_SD2 | Uncontrolled | Object de-orbit | Space debris |

*3.2. Use Cases in the European Context*

Not all use cases listed in Table 1—such as controlled satellite de-orbits—are likely to occur in the European Network Area to the same extent.

On the other hand, at least some among the use cases are operative for years or decades, such as stratospheric balloons. Supersonic aircraft were operated in the past and could have a comeback in the future, whereas other use cases, such as orbital rocket launches, are common in other parts of the world and are foreseen to be operated in the European region regularly as well, though in a much smaller scale in the form of so-called mini- and micro-launchers. It is to be expected that both expendable and—as technology progresses—reusable rocket systems may be operated.

Due to different technological readiness levels—some use cases are concepts only at this point in time—and the resulting high degree of uncertainty regarding the timing of implementation, the developed scenarios incorporate current available information and expertise. For the same reason, only use cases that are already operational, regarded to be realistic in Europe, and/or have documented operational intentions at the time of the study were considered for the impact analysis [9]. The four European regional scenarios are Scandinavia, the United Kingdom (UK)–Ireland functional airspace block (FAB), the functional airspace block of Central Europe (FABEC), and the (East) Mediterranean area. These use cases are listed in Table 2.

**Table 2.** HAO Use Cases considered per region.

| Use Case | Identifier | Scandinavia | UK-Ireland FAB | FABEC | (East) Mediterranean |
|---|---|---|---|---|---|
| HAPS | UC_HAPS_FB1 | X | X | X | X |
| | UC_HAPS_AS1 | X | X | X | X |
| | UC_HAPS_AC1 | X | X | X | X |
| Launchers | UC_LAUN_SR1 | X | X | X | X |
| | UC_LAUN_DE1 | X | X | | |
| | UC_LAUN_DR1 | X | X | | |
| | UC_LAUN_AE1 | X | X | | |
| A-to-A | UC_ATOA_AL1 | | X | | X |
| | UC_ATOA_VR1 | | | | X |
| A-to-B | UC_ATOB_SA1 | X | X | X | X |
| | UC_ATOB_HA1 | | X | | |
| From Orbit | UC_FORB_RV1 | | | X | X |

## 4. Demand Scenarios

Instead of creating one large and overwhelming scenario framework for the European Network region, four regional scenarios (see Table 2) were developed. Out of these four, the UK–Ireland FAB regional scenario is the focus of the impact assessment in this paper.

### 4.1. Regional Scenario Principles

A wide range of possible real-world conditions and environments have been considered and should be covered. The four areas that were introduced each feature a unique combination of key characteristics, which are geographical location, population density, fragmentation pressure, and air traffic density [9].

These characteristics were defined as follows:

- Geographical location:
  Here, the geographical latitude of the spaceport or launch site plays an important role since it is the limiting factor of the operational windows for some types of operations, such as, for example, HAPS during the winter season. The four areas cover the whole latitude range of the European Network area.
- Population density:
  Population density serves as a measurement of the number of people per unit land area. For example, a low population density is in two ways beneficial for new entrants' operations. First, it is beneficial in terms of the feasibility of immature vehicle operations, and secondly, it is beneficial because operations such as HAPS are expected to provide services in sparsely populated areas.
- Fragmentation pressure:
  This is a measure of the degree to which movement between different parts of a landscape is interrupted by fragmentation geometry (for example, streets or other infrastructure). This adds another layer of information by which to identify remote areas. The two metrics of fragmentation pressure and population density enable us to obtain an informative overview of potentially relevant regions for HAO within the European Network Area.
- Air traffic density:
  This characteristic is represented by the line density of the flight tracks from 28 June 2019, which is the busiest traffic day in European airspace to date, within the respective area.

The four areas subject to investigation are as follows. If applicable, the lower airspace boundaries of the Flight Information Regions (FIR) were used in order to increase granularity.

- Scandinavia:
  This region is comprised of the Bodo Oceanic FIR, Polaris FIR, Koebenhavn FIR, Sweden FIR, and Finland FIR. This area roughly covers the combined areas of the Danish–Swedish FAB and the North European FAB.
- United Kingdom–Ireland FAB:
  This region is comprised of the Shannon FIR, London FIR, and Scottish FIR.
- FABEC:
  This region is comprised of the Brest FIR, Bordeaux FIR, Paris FIR, Marseille FIR, Reims FIR, Switzerland FIR, Brussels FIR, Amsterdam FIR, Langen FIR, and Bremen FIR.
- (East) Mediterranean:
  This region is comprised of the Roma FIR, Malta FIR, Brindisi FIR, and Athinai FIR. This area is a sub-area of the Blue Med FAB.

As shown below in Table 3, the four areas and their characteristics represent a representative cross-section of real-world conditions that HAO will face in the European Network Area once deployed.

**Table 3.** Areas covered by demand scenarios and key characteristics.

| Regional Scenario | Latitude | Population Density & Fragmentation | Qualitative Valuation of (Current) Air Traffic Density |
|---|---|---|---|
| Scandinavia | High (up to high 60 s) | Predominantly low fragmentation over sparsely populated areas | Low |
| UK–Ireland FAB | Middle to high (50° to low 60 s) | Approx. 25% of land with low fragmentation over sparsely populated areas (Northern Scotland); otherwise, medium to high fragmentation over medium to densely populated areas | High |
| FABEC | Middle (40° to mid 50 s) | High fragmentation over medium to high population density areas | High |
| (East) Mediterranean | Low to middle (mid 30 s to low 40 s) | Average fragmentation over medium population density areas. | Medium |

*4.2. Scenario Timescale*

Three different time horizons were used as reference points for the HAO demand scenarios: short-, medium-, and long-term. The demand scenario traffic numbers roughly covered the time spans 2025–2030, 2030–2035, and 2035+, respectively. As uncertainty dominates, especially in the more distant scenarios, the main objective of the demand scenarios is to outline a plausible and coherent evolution of future HAO activities. Accordingly, there are, in total, 12 HAO demand scenarios, three per each region (I/II/III) [9].

From an air traffic management (ATM) point of view, managing HAOs will presumably demand an incremental approach. Starting from the status quo, a certain point in time must be anticipated where new ATM services, capabilities, or regulations are needed in order to keep up with the growing demand. In this study, this point is referred to as T1. A second transition point, T2, marks the transfer to a new scalable form of management. Generally speaking, T1 represents the start of an incremental improvement phase, whereas T2 marks the launch of the "new" regime of (higher airspace) ATM. This approach is called "time as an output", contrasting "time as an input" [9].

It is important to note that both approaches—the short-/medium-/long-term on the one hand and T1/T2 on the other hand—were assessed and found to be best applicable in different contexts. The first is an integral and primary reference of the demand forecast, indicating the evolution of demand over time. The latter is a qualitative approach and used hereafter to create progressively challenging scenarios with regard to the ATM system. This decision is supported with respect to the pursuit of measuring and evaluating the impact

of HAOs on the network. The demand numbers are nevertheless scrutinized to be coherent and plausible based on the information available [9].

*4.3. Demand Assumptions*

The demand scenarios, despite being created under the condition of time as an output, require robust and sound demand data. In this regard, assumptions about conditions and operational requirements for viable HAOs were formulated. The assumptions used for the regional demand scenarios are presented in the following sub-sections below.

### 4.3.1. HAPS

Motorized HAPS are expected to provide predominantly telecommunication services in areas with low to medium population density. In addition, HAPS can be used, among other things, for maritime surveillance and border security measures.

Due to limited maneuverability and low speeds, the transition through lower and upper airspace is a critical flight phase for HAPS. Blocking large airspaces for a considerable amount of time must be assessed regarding its potential impact on the network (and will be addressed in the impact assessment). The maximum number of expected HAPS transitions will therefore be explicitly listed.

Regarding the four regional scenarios, the following areas are considered as realistic for continuous operations (considering operational aspects and the utilization of HAPS) [9]:

- Scandinavia:
  All parts due to low population density, except the southern parts of Sweden and Finland, though wintry conditions could pose notable operational challenges.
- UK–Ireland FAB:
  The northern parts of England and Scotland and western and southern Ireland.
- FABEC:
  Continuous operations are not likely over mainland (medium-high population density). Maritime coverage and special missions and/or test flights are realistic, though.
- (East) Mediterranean:
  Sardinia, southern Italy, Greece, and maritime coverage are realistic.

### 4.3.2. Sub-Orbital

Currently, frequently operated sub-orbital operations in the European Network area are taking place in the form of sounding rocket launches (almost exclusively in Scandinavia). Future demand is expected for the UK as well. A-to-A flights may be launched in the UK and Italy.

Supersonic A-to-B flights could be revived, though they would only connect major city pairs. Within the four European regions investigated, the following cities have been identified as most likely destinations [9]. They are presented in Table 4:

**Table 4.** Identified (Hyper-) Supersonic Destinations in Europe.

| Region | City |
|---|---|
| Scandinavia | - |
| UK-Ireland FAB | London |
| FABEC | Paris, Amsterdam, Frankfurt, Munich, Zurich |
| (East) Mediterranean | Rome |

### 4.3.3. Launchers

Orbital launch activities are expected to emerge in Scandinavia and the UK–Ireland FAB area. Possible launch sites/spaceports have been identified, with initial launch intentions announced or, in fact, already materialized [9].

### 4.3.4. From Orbit

The from-orbit operations of Space Rider may use Grottaglie in southern Italy as its landing site, which would affect both the southern FABEC area and the investigated area of the (East) Mediterranean. Dream Chaser approaches into Cornwall in the southwest of the UK are under consideration as well [9].

### 4.4. Scenario: UK–Ireland FAB

The following launch and/or take-off sites regarding operations concerning the higher airspace environment were identified and considered for the demand scenarios [9]. They are detailed in Table 5. Additionally, London Heathrow airport was added as a starting point for hypersonic A-to-B operations.

It should be noted that the identifiers of the spaceports and launch sites are not consecutively numbered at all times due to the subsequent appendment of additional sites. Renaming was considered to be impracticable because the simulation implementation had already started at this stage.

**Table 5.** HAO Scenario for the UK–Ireland FAB: launch sites/spaceports.

| Identifier | Launch Site/Spaceport | Territory | Coordinates |
|---|---|---|---|
| 00 | London Heathrow | United Kingdom | 51.48, −0.46 |
| 04 | SaxaVord Spaceport | United Kingdom | 60.82, −0.77 |
| 05 | Space Hub Sutherland | United Kingdom | 58.51, −4.51 |
| 06 | Spaceport 1 | United Kingdom | 57.65, −7.49 |
| 07 | Spaceport Macrihanish | United Kingdom | 55.44, −5.69 |
| 10 | Spaceport Cornwall | United Kingdom | 50.44, −4.99 |
| 26 | Kilkenny Airport [1] | Ireland | 52.65, −7.30 |

[1] No underlying information; notional scenario-specific HAPS launch site.

Based on the UK–Ireland region, the scenario numbers were allocated to three consecutive demand scenarios (see Section 4.2). The demand estimates rise with the progression of time, corresponding with the advent of new technology and other presumably occurring operational enablers.

Due to the analysis being performed on a regional level, specifically for the UK–Ireland FAB in the context of this report, certain trans-regional or global operations, such as supersonic A-to-B flights, can be characterized as departing, arriving, or crossing traffic in a given region. The terms are defined as follows:

- Departure refers to an operation originating from the region with the intent to land at a destination in a different region from that of its departure. An arrival, accordingly, is an operation landing in a region that had a departure region different from its destination.
- A crossing with respect to the region refers to a trans-regional operation that is neither departing nor arriving in the region but is using the airspace of the region on its route.

Finally, the categories of the respective demand scenarios in Table 6 were created based on criteria developed during the ECHO project [9]:

- Frequency of scheduled hyper- (HS) and supersonic (SS) flights:
  Operations are supposed to be scheduled for both supersonic (SS) and hypersonic (HS) flights in correspondence with the flight schedules of conventional mainline airline operations. Therefore, the figure for the daily number of departures and arrivals represents both departures and arrivals.
- Business use of hyper- and supersonic flights:
  Business use is usually comprised of on-demand operations. The figure stated is therefore the sum of the departures and arrivals.
- Continuous HAPS operations:
  The total number of HAPS operating at the same time.

- HAPS maximum transitions:
  The maximum number of HAPS that is expected to transition from ground to higher airspace or vice versa on a single day.
- Balloon–HAPS operations:
  The total number of balloon–HAPS expected to launch in a year.
- Sounding rocket launches:
  The total number of sounding rockets expected to launch in a year.
- Other sub-orbital launches:
  The total number of other sub-orbital flights, such as touristic A-to-A experiences, expected to launch in a year.
- Orbital operations:
  The total number of orbital operations, whether launches or from-orbit operations, expected in a year.

**Table 6.** HAO scenario for the UK–Ireland FAB: demand.

| Catagory | Demand Scenario I | Demand Scenario II | Demand Scenario III |
|---|---|---|---|
| Frequency of scheduled hyper- (HS) and supersonic (SS) flights | Dep./arr.: 0-1/day (SS) | Crossings: 6–8/day (SS) Dep./arr.: 2–3/day (SS) | Crossings: 16–24/day (SS) Dep./arr.: 4–8/day (SS) 0–1/day (HS) |
| Business use of hyper- and supersonic flights | 0 | Crossings: 2–3/day (SS) Dep. or arr.: 2–3/day (SS) | Crossings: 8–12/day Dep. or arr.: 4–6/day |
| Continuous HAPS operations | 1–2 | 10–20 | 20–30 |
| HAPS max. transitions | 1/day | 1–2/day | 2/day |
| Balloon–HAPS operations | 0 | 0–6/year | 0–6/year |
| Sounding rocket launches | 12–18/year | 12–18/year | 12–18/year |
| Other sub-orbital launches | 2–4 | 12/year | 25–30/year |
| Orbital operations | 4–8/year | 20–25/year | 40–60/year |

## 5. Impact Assessment

In this section, the impact assessment of HAOs on conventional air traffic is described. Thereby, the findings of Sections 3 and 4 are used as input to the impact assessment, which was performed with the help of simulations. Since the UK–Ireland region has a leading role in the implementation of HAOs and commercial space activities, the impact assessment focused upon this region. Additionally, there is a wealth of information available for this region, which is a major requirement regarding the modeling of HAOs within a simulation. Another reason for the focus on UK–Ireland is the large variety of use cases, which is not present to this extent in the other three regions (see Table 2). The synthesis of new entrants did not identify from-orbit operations as likely for UK–Ireland. To represent each of the five main categories of vehicles within the impact assessment, from-orbit operations will be modeled exceptionally in the UK–Ireland region.

In this section, first the general approach of the assessment is illustrated, and then, the simulation setup is outlined in more detail.

### 5.1. General Approach

This impact analysis focused on an assessment of the potential impact of different types of HAOs on the existing air traffic, specifically considering interactions with air

traffic in airspaces below flight level (FL) 660. To assess this impact, a fast-time simulation was performed with the tool AirTOP. Being a widely used fast-time simulation software, AirTOP is able to perform gate-to-gate air traffic simulations and is capable of supporting relevant en route structures and controller tasks [10]. This allows an overall assessment of air traffic as well as a simulation of ATM actions, such as airspace closures and reroutings. Consequently, AirTOP is an appropriate tool for the impact analysis.

In order to quantify the impact of HAOs on conventional air traffic, two types of simulations were conducted:

- A static simulation;
- A dynamic simulation.

The aim of the static simulation was to log the conventional aircraft that were affected by the use cases (see Section 3.1). One day of air traffic operations was simulated, and the amount of operational interactions between conventional air traffic and vehicle use cases were measured during the simulation runs. Additionally, the results of the static simulation served as a baseline because the air traffic did not have to avoid interactions with the HAOs. The results were then compared to the results of the dynamic simulation.

Regarding the dynamic simulation, the conventional air traffic was supposed to avoid interactions with HAOs. The aim was to measure global effects per demand scenario from the avoidance of interactions based on an operational scenario. HAOs only occurred during scheduled intervals, which were determined by means of operational characteristics, and only one demand scenario was simulated at once so that the effects could be quantified for each scenario.

The impact of HAOs on conventional air traffic was measured by means of the following flight efficiency parameters:

- Total flight distance of each aircraft in the area of interest (AOI):
  This parameter was determined for each simulation type and compared between the static simulation and the dynamic simulation to quantify changes.
- Total flight duration of each aircraft in the AOI:
  This parameter was determined for each simulation type and compared between the static simulation and the dynamic simulation to quantify changes.
- Total fuel consumption of each aircraft in the AOI:
  This parameter was determined for each simulation type and compared between the static simulation and the dynamic simulation to quantify changes.
- Number of interactions between aircraft and HAOs:
  This parameter was determined for the static simulation.
- Number of reroutings being performed because of HAOs:
  This parameter was determined for the dynamic simulation and compared to the number of interactions between aircraft and HAOs.

### 5.2. Simulation Setup

To assess the impact of HAOs on existing air traffic, interactions between the vehicle use cases and conventional air traffic were analyzed. For this purpose, the simulation model had to represent the vehicle use cases (see Section 3.1). Since there are different categories of HAVs, a large variety of performances exists [11], and the performance of HAVs differs notably from conventional aircraft performance. However, AirTOP is capable of simulating flight movements but is limited to the performance characteristics of aircraft that operate below FL660 with subsonic speeds. Because of this, the specific motions of the HAVs and their resulting trajectories were not modeled. Instead, it was assumed that they operated inside the boundaries of polyhedrons, the dimensions of which were equal to the surrounding restricted airspaces. This modeling technique of non-aircraft-like operations was based on the study by Young, Kee, and Young [3].

### 5.2.1. Modeling of HAO Polyhedrons

Below, the polyhedron shapes for the five main categories of vehicles are described, namely for the HAPS, A-to-A, A-to-B, launcher, and from-orbit aircraft.

HAPS

The dimensions of polyhedrons for HAPS were inspired by the Airbus Zephyr operations at Wyndham Airfield in Australia [12]. They were not exactly identical since operations of experimental air vehicles outside European airspace are usually designed with generous protected areas, so smaller dimensions were assumed to be practicable for European airspace. Given the fact that HAPS operate at a mission altitude above FL550 [12] where conventional air traffic does not occur, only the ascent and descent of HAPS were modeled in the simulation. Table 7 shows the dimensions of the polyhedrons that were used for HAPS.

**Table 7.** Dimensions of polyhedrons for HAPS.

| Segment | Radius (in NM) |
|---------|----------------|
| Ground–FL50 | 5 |
| FL50–FL100 | 10 |
| FL100–FL200 | 25 |
| FL200–FL550 | 75 (divided into six sub-zones, each of 60°) |

The upper segment was divided into six sub-zones, but only one sub-zone at a time was needed by the vehicle. An exemplary shape of HAPS polyhedrons in the simulation model is shown in Figure 1. It represents the vertical transition zone for ascent and descent.

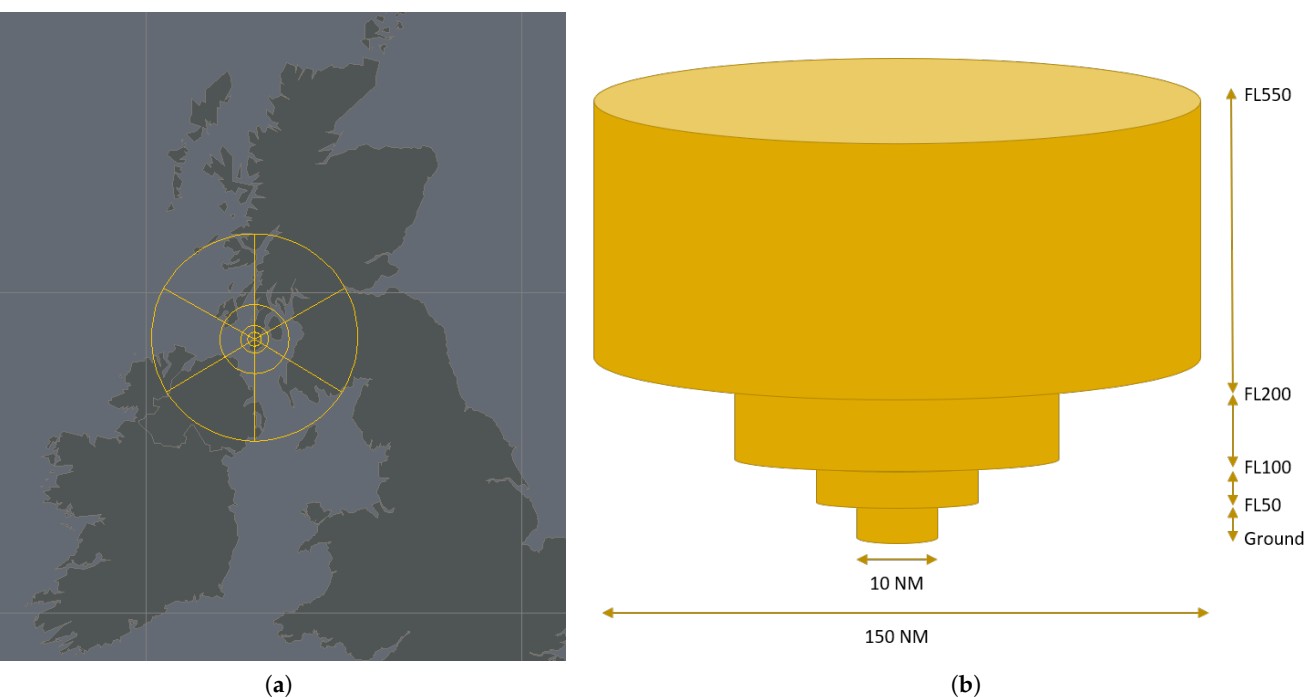

**Figure 1.** HAPS polyhedrons: (**a**) top view; (**b**) vertical classification.

Launchers

The simulation featured two types of use cases for launchers, namely air launches and direct launches. The polyhedrons of air launches (UC_LAUN_AE1) were based on planned Virgin Orbit operations (as of 11 March 2022) from the Cornwall spaceport [13]. Simulations were performed using this level of knowledge since the operations at the

Cornwall spaceport were in the planning phase at that time. The actual launch on 9 January 2023 featured these temporary danger areas (TDA) plus additional restricted airspaces [14]. Since this was the first launch of such a type of operation, it seems likely that the amount of TDAs can be reduced to the polyhedrons modeled in the present study if similar operations will be carried out in the future.

The vertical dimension of the polyhedrons extended from ground level to infinity. Per this operation, this use case consisted of two polyhedrons, which modeled the TDAs for rocket ignition and the splashdown of the first stage. The shape of the polyhedrons used for air launches can be seen in Figure 2a.

The polyhedrons of direct launches (UC_LAUN_DE1) were based on the intended HyImpulse sounding rocket launch from SaxaVord Spaceport–Shetland (as of 11 May 2022) [15], and the necessary drop zones were based on drop zones from the Rocket Lab Electron launch on 15 May 2021, which were obtained through a related notice to airmen (NOTAM) from a Federal Aviation Administration website [16]. The vertical dimension of the polyhedrons extended from ground level to infinity. The shape of the polyhedrons used for direct launches is illustrated in Figure 2b.

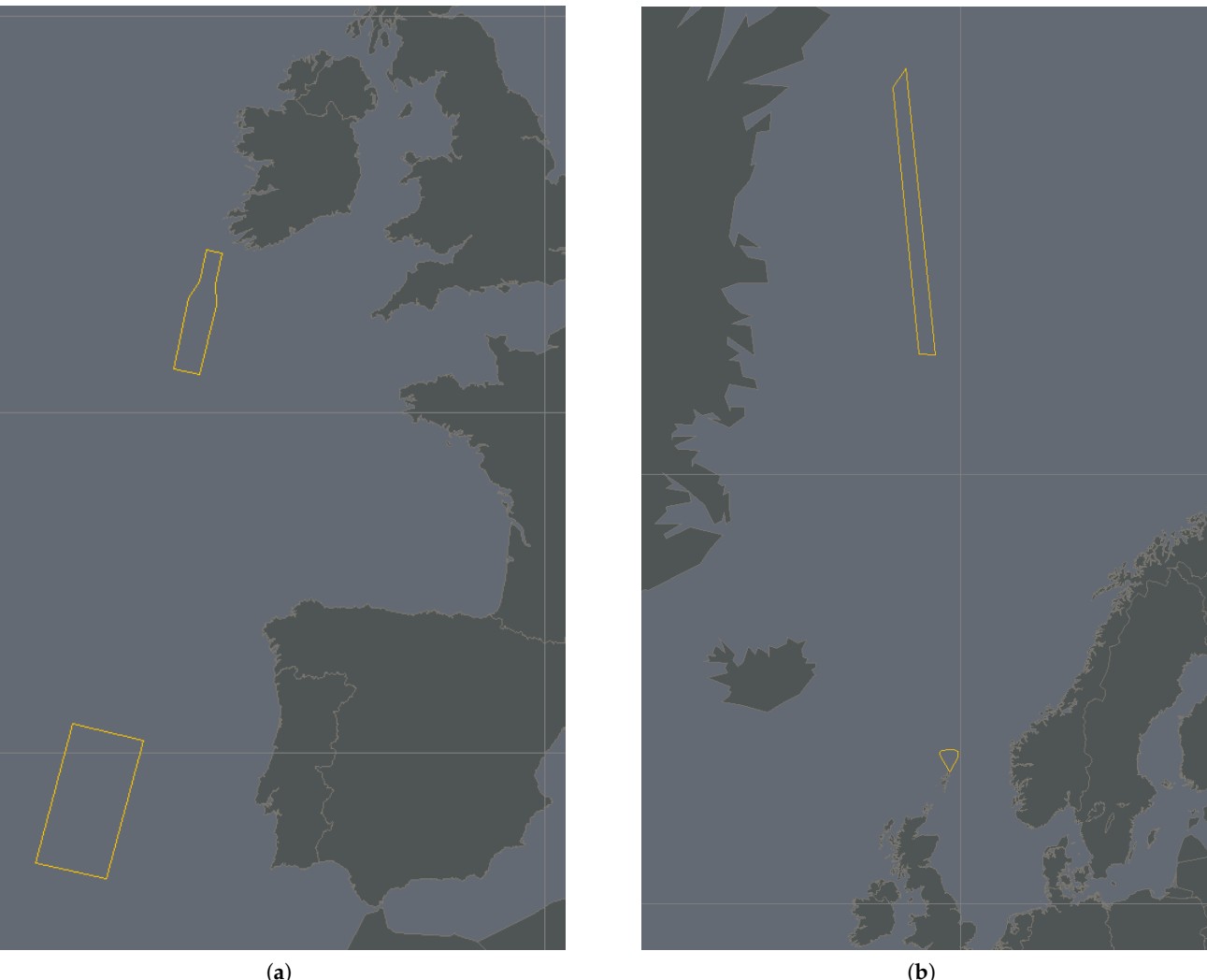

(**a**)                                   (**b**)

**Figure 2.** Launchers' polyhedrons: (**a**) air launch polyhedrons; (**b**) direct launch polyhedrons.

At Spaceport 1, the required dimensions of polyhedrons for direct launches were directly derived by the corresponding airspace change request (as of 9 March 2022). The TDAs around the launch site covered parts of the existing Hebrides Range Danger Area D701, mainly D701C, D701E, and D701F [17]. Necessary drop zones were based on the drop

zones from the Rocket Lab Electron launch on 15 May 2021, which were obtained through related NOTAMs [16]. The vertical dimension of the polyhedrons extended from ground level to infinity. Figure 3 shows the shape of the polyhedrons used for direct launches at Spaceport 1.

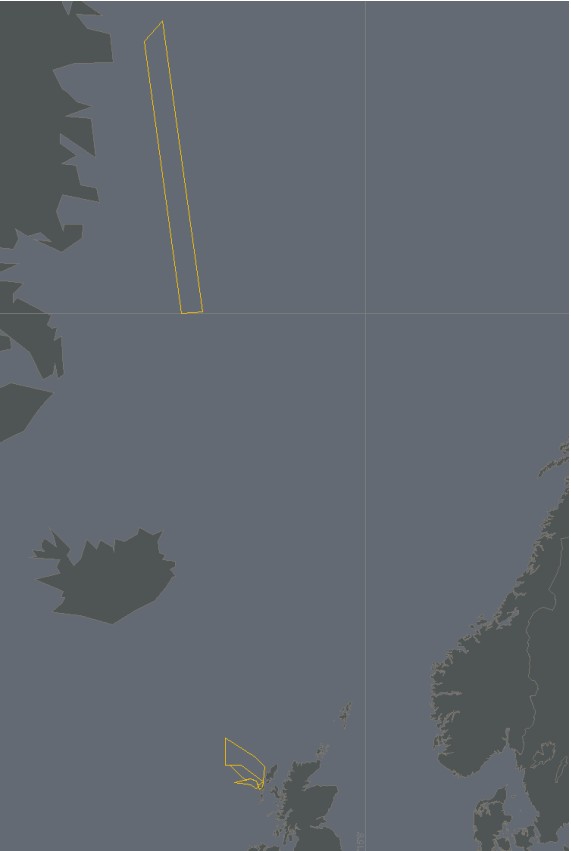

**Figure 3.** Direct launch polyhedrons at Spaceport 1.

A-to-A

The polyhedrons of sub-orbital A-to-A operations (UC_ATOA_AL1) were based on the restricted airspace from the Virgin Galactic flight on 11 July 2021, which was obtained through related NOTAMs [16]. The vertical dimension of the polyhedrons extended from ground level to infinity. An exemplary shape of a polyhedron for sub-orbital A-to-A operations is presented in Figure 4.

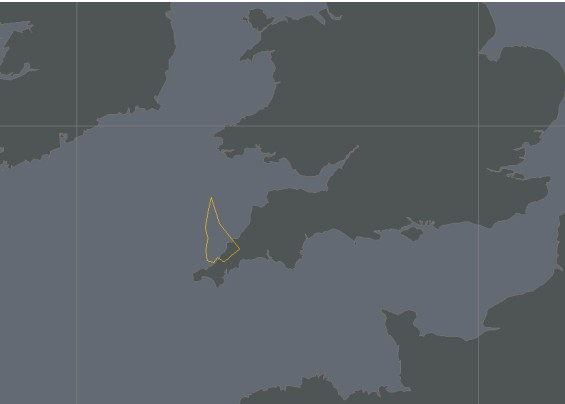

**Figure 4.** A-to-A sub-orbital polyhedron.

A-to-B

As one result of the ECHO project, hypersonic A-to-B operations (UC_ATOB_HA1) require a segregated area during their hypersonic ascent and descent. Such a segregated area for UC_ATOB_HA1 was assumed to be a polyhedron of a rectangular form with a width of 10 NM and a length of 100 NM. The width originated from Par. 1.1.2 of the UK Aeronautical Information Publication, part ENR1.1 [18], where it is stated that the width of an airway is 5 NM on either side of a straight line, which equals a total width of 10 NM. The hypersonic ascent and descent was expected to cover a long distance, so a length of 100 NM was set. The vertical dimension of the polyhedrons extended from FL360 to FL660 because the hypersonic ascent and descent were assumed to take place between these flight levels. A model of this use case can be seen in Figure 5.

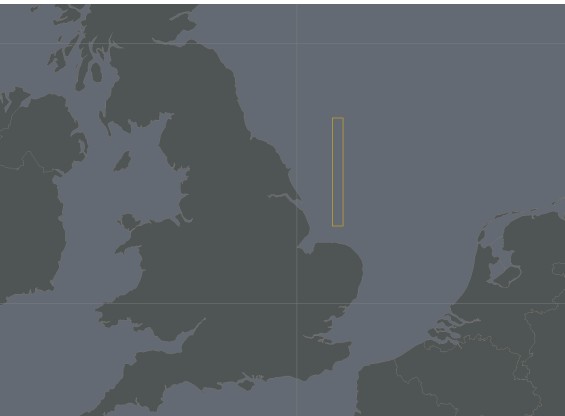

**Figure 5.** A-to-B polyhedron.

From Orbit

Regarding from-orbit operations, the polyhedrons of this use case (UC_FORB_RV1) were based on the Sierra Space Dream Chaser usage [19]. The shape was set to be a circle segment of 135° with a radius of 20 NM. The vertical dimension of the polyhedrons extended from ground level to FL660 because this altitude range is of interest concerning the impact on the existing air traffic. Figure 6 presents the shape of polyhedrons for from-orbit operations.

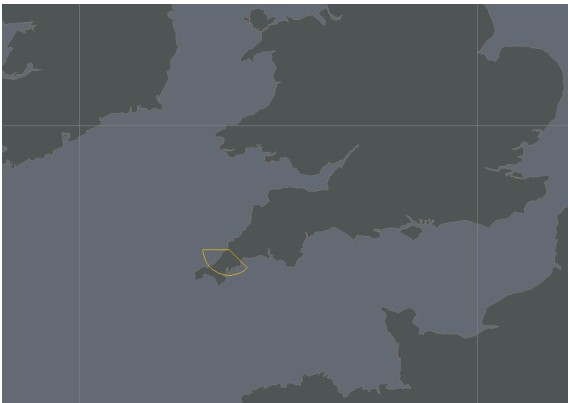

**Figure 6.** From-orbit polyhedron.

As reported above, the use cases of HAPS and launchers consisted of multiple polyhedrons. Regarding HAPS, the upper segment was divided into six sub-zones, each of 60° (see Table 7). As a vehicle needed only one sub-zone at a time, a selection of an active sub-zone per HAPS use case was made for the dynamic simulation. Regarding launchers, a selection of active polyhedrons was made as well.

An overview of active use cases per each demand scenario that was covered in the simulation is presented in Figure 7.

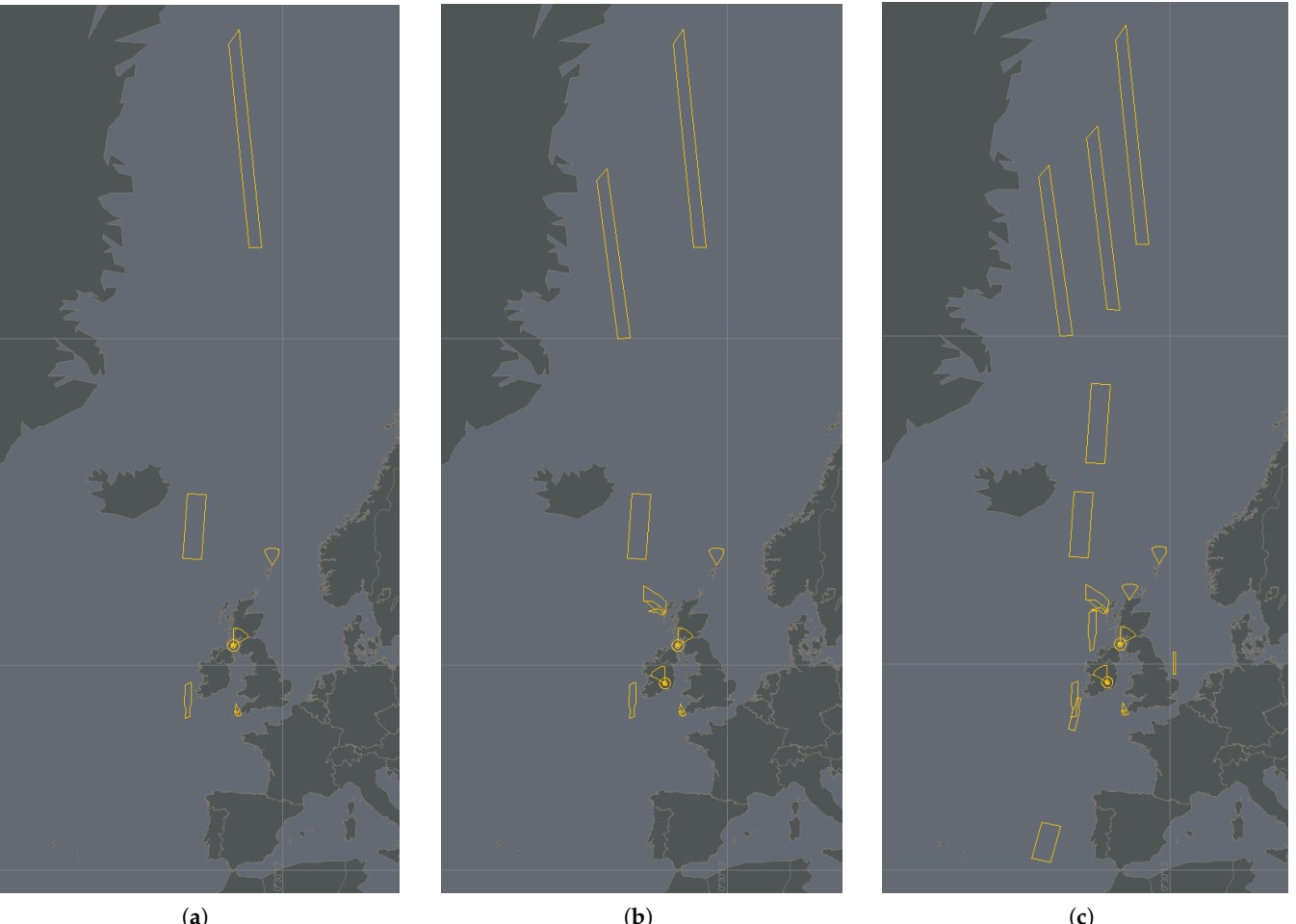

**Figure 7.** Overview of use case polyhedrons covered in the simulation per demand scenario: (**a**) demand scenario I; (**b**) demand scenario II; (**c**) demand scenario III.

5.2.2. Airspace and Flight Plan

The airspace modeled in the fast-time simulation was a so-called AOI that consisted of the FIRs of the regional scenarios for Scandinavia, UK–Ireland, FABEC, and the (East) Mediterranean (see Section 4.1), as well as bordering FIRs that include parts of vehicle use cases. The following FIRs were included fully or partially in the simulation model, indicated as per their International Civil Aviation Organization (ICAO) code: ENOB, ENOR, ESAA, EKDK, EFIN, EGPX, EISN, EGTT, EDUU, EDVV, EHAA, EBUR, LSAS, LFFF, LIBB, LIRR, LMMM, LGGG, BIRD, BGGL, LPPC, LECB, LECM, EGGX, and LPPO. The coordinates of each FIR were obtained through EUROCONTROL's Aviation Data for Research Repository [20] for the year 2019. The northern, southern, and eastern borders of the AOI were equivalent to the borders of the FIRs; the western border was placed at 30°W longitude to include all areas of HAOs.

Regarding the modeling of the conventional air traffic, the day with the most recorded flights of the year 2019 (28 June 2019) in the European Civil Aviation Conference area was chosen as a flight plan for the impact assessment. The flight plan data of this day were obtained from the Demand Data Repository (DDR2) by EUROCONTROL [21]. The flight plans were based on the last filed flight plan (filed tactical flight model) from the operators. The traffic sample of this day included flights that departed, arrived, or transited the AOI. The simulation injection times were departure times at the respective airports, and the

simulated trajectories were based on aircraft performance data derived from the Base of Aircraft Data by EUROCONTROL (version 3) [22]. The flight plan data were modified by the AOI as follows:

- Case 1: Origin and destination inside AOI → flight was simulated from origin to destination;
- Case 2: Origin in AOI, destination outside AOI → flight was simulated from origin to the first waypoint outside the AOI;
- Case 3: Origin outside AOI, destination inside AOI → flight was simulated from the last waypoint outside the AOI to the destination;
- Case 4: Origin and destination outside the AOI → flight was simulated only inside the AOI (between the entry and exit waypoints).

In conclusion, within each case, a flight had a starting point and an endpoint that were case-specific. Since one single day of operation was considered, specific effects on this day were included within the filed flight plan trajectories from the DDR2. These effects included the runway direction of airports on this day, which implies the inclusion of standard instrument departure routes and standard arrival routes, as well as airspace closures. The focus of this simulation was on en route traffic, so airport-related effects were an undesired impact factor. Airspace closures were undesired factors as well because they can vary on a daily basis. To exclude these effects within the flight plan trajectories filed from the DDR2, all flights were routed along the smallest circular distance between their starting point and endpoint in the simulation.

5.2.3. Intervals for Use Cases

The dynamic simulation considered scheduled intervals for the polyhedrons of the use cases, which were determined on the basis of operational characteristics and specificities. Regarding HAPS, the considered use cases were from the type "UC_HAPS_AC1", which were Zephyr-like operations. Therefore, the operational characteristics were derived from this type. The Airbus Zephyr has an ascent/descent rate of 100 ft/min [12] that leads to the ascent/descent duration presented in Table 8.

**Table 8.** Duration for ascent/descent UC_HAPS_AC1.

| Segment | Duration for Ascent/Descent (in min) |
|---|---|
| Ground–FL50 | 50 |
| FL50–FL100 | 50 |
| FL100–FL200 | 100 |
| FL200–FL550 | 350 |
| Ground–FL550 | 550 |

The subsequent upper segment was opened 15 min before the closing time of the lower segment. It was assumed that HAPS operations had an ascent window that started at dawn. The dynamic simulation only assessed the ascent of HAPS operations since the descent ends at dawn and therefore occurs during the night, which leads to many fewer interactions with the conventional air traffic.

Concerning launcher, A-to-A, and from-orbit operations, a time window of one hour was assumed to be necessary for operation. Operations could occur at any time. The duration of the time window regarding the segregated area for the hypersonic ascent/descent of hypersonic A-to-B operations was expected to be 30 min, and operations could occur at any time.

Based on these operational characteristics, an analysis was performed to identify scheduled intervals for the operation of each use case in UK–Ireland. Within this analysis, the time window of each use case's operation was shifted every 10 min between 0:00 a.m. and 11:00 p.m. to count the interactions between the conventional air traffic and the respective use case during the operational duration. The identified scheduled intervals (in

UTC time) for each use case are listed in Table 9. The notation of the use cases is combined with the regional identifier (number "2" for UK–Ireland), the launch site/spaceport identifier (see Table 5), and the use case type identifier (see Table 2). The number in front of the time stamp specifies the day (2 = 28 June 2019). The scheduled intervals were identified on the basis of minimizing the interactions between air traffic and HAOs. Additionally, interactions between the use cases were excluded. HAPS operations started at dawn in UK–Ireland, and hypersonic A-to-B operations considered a mission to Australia with respect to the time difference.

**Table 9.** Scheduled intervals for use cases in the dynamic simulation.

| Use Case | Scheduled Interval |
| --- | --- |
| 2_07_UC_HAPS_AC1 | 2 03:40:00–2 12:49:59 |
| 2_26_UC_HAPS_AC1 | 2 04:00:00–2 13:09:59 |
| 2_04_UC_LAUN_DE1 | 2 07:00:00–2 07:59:59 |
| 2_05_UC_LAUN_DE1 | 2 06:00:00–2 06:59:59 |
| 2_06_UC_LAUN_DE1 | 2 08:00:00–2 08:59:59 |
| 2_07_UC_LAUN_AE1 | 2 19:40:00–2 20:39:59 |
| 2_10_UC_LAUN_AE1 (Northern trajectory) | 2 22:30:00–2 23:29:59 |
| 2_10_UC_LAUN_AE1 (Southern trajectory) | 2 20:20:00–2 21:19:59 |
| 2_10_UC_ATOA_AL1 | 2 16:30:00–2 17:29:59 |
| 2_10_UC_FORB_RV1 | 2 04:00:00–2 04:59:59 |
| 2_00_UC_ATOB_HA1 | 2 21:30:00–2 21:59:59 |

Within these intervals, conventional aircraft were prohibited to fly through the respective polyhedrons, which means that rerouting was necessary. A visualization of the rerouting process can be seen in Figure 8. The two most relevant parameter settings are the rerouting margin and the distance to start/stop the rerouting. In the case of the rerouting margin, this parameter defines the lateral spacing between the aircraft and polyhedron (airspace to avoid). AirTOP requires a value greater than 0 NM, so a margin of 1 NM was set in the simulation since the dimensions of the polyhedrons already covered the necessary restricted airspaces for aircraft. The distance to start and stop the rerouting is the longitudinal distance at which the aircraft begins or ends its rerouting around the polyhedron (airspace to avoid). This parameter was set at 50 NM to avoid unrealistic, sharp aircraft movements. AirTOP identified flights that intersected polyhedrons during the scheduled interval (see Table 9). These flights were routed around the polyhedrons by the shortest route while respecting the aforementioned parameter settings. Therefore, the extra distance flown was minimized.

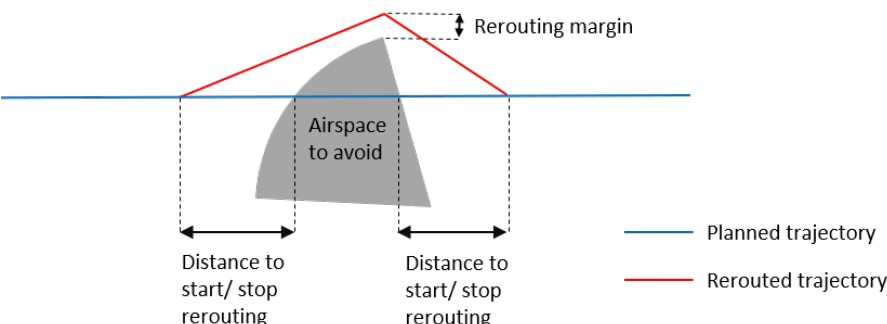

**Figure 8.** Rerouting process in AirTOP.

5.2.4. Assumptions and Limitations

Below, a list of assumptions and limitations regarding the simulation and impact analysis is addressed.

- The original flight plan data were edited so that round trips (flights where the origin and destination were identical) and military flights were excluded from the simulation.

- In the dynamic simulation, flights only performed rerouting maneuvers as a result of closed airspaces from HAV operations.
- Airports and related terminal maneuvering areas were not modeled. Therefore, impacts related to airport operations could not be determined.
- ATC sectors and controllers were not modeled in detail.
- Controller workload was not considered in the simulation, so each flight that could be rerouted was rerouted automatically.
- No actions were performed by the simulator to separate aircraft.
- Actual trajectories of the HAVs were not modeled; it was assumed that they were within their polyhedrons.
- The movement of carrier aircraft to their launch area for air launches was not modeled; it was assumed that this mission part can be integrated into the air traffic flow conventionally.
- Weather events were not included in the simulation.
- Flights could not be rerouted if their starting point or endpoint were inside a closed airspace.
- Off-nominal events of HAVs, such as structural or system failures, were not the focus of the impact analysis.

## 6. Results

In this section, the principal results of the simulation runs are described and presented on the basis of the output parameters described in Section 5.1.

### 6.1. Static Simulation

Regarding the static simulation, the total number of aircraft flying through the polyhedrons of a use case, which symbolizes the number of interactions between the conventional air traffic and HAOs, were determined. The interactions were measured for the entire day of 28 June 2019. Table 10 lists the results of this output parameter per use case.

**Table 10.** Total number of interactions with conventional air traffic per use case.

| Use Case | Total Number of Interactions with Conventional Air Traffic per Use Case |
|---|---|
| 2_00_UC_ATOB_HA1 | 338 |
| 2_04_UC_LAUN_DE1 | 38 |
| 2_05_UC_LAUN_DE1 | 66 |
| 2_06_UC_LAUN_DE1 | 205 |
| 2_07_UC_HAPS_AC1 | 1644 |
| 2_07_UC_LAUN_AE1 | 411 |
| 2_10_UC_ATOA_AL1 | 178 |
| 2_10_UC_FORB_RV1 | 83 |
| 2_10_UC_LAUN_AE1 (Northern trajectory) | 420 |
| 2_10_UC_LAUN_AE1 (Southern trajectory) | 310 |
| 2_26_UC_HAPS_AC1 | 2494 |

The content of Table 10 can be aggregated for each of the five main categories of the vehicle use cases, which is presented in Table 11. It can be seen that HAPS caused the most interactions with conventional air traffic. Launchers experienced the second greatest number of interactions with the air traffic but notably lower interactions than HAPS. ATOB operations had 160 more interactions than ATOA operations. The fewest interactions occurred with FORB operations.

**Table 11.** Total number of interactions with conventional air traffic per use case category.

| Use Case | Total Number of Interactions with Conventional Air Traffic per Use Case Category |
|---|---|
| HAPS | 4138 |
| LAUN | 1450 |
| ATOA | 178 |
| ATOB | 338 |
| FORB | 83 |

*6.2. Dynamic Simulation*

In our next step, the results of the dynamic simulation for UK–Ireland are considered. It was mentioned in Section 5.1 that the dynamic simulation was solely conducted for the traffic level for one day, but each of the three demand scenarios was considered. Therefore, three specific simulation runs were performed. Within the dynamic simulation, changes in the flight efficiency parameters, specifically the total flight distance, total flight duration, and total fuel consumption, due to conventional air traffic were analyzed. Therefore, the results obtained with these parameters in the dynamic simulation were compared against the results from the static simulation. The statistical significance of these changes was examined with a paired *t*-test. Furthermore, the number of reroutings that were performed because of polyhedrons of the use cases was determined.

The changes in the flight efficiency parameters, which were compared to the static simulation, were measured only for aircraft that had to reroute around polyhedrons since there were no other interactions than reroutings that could have affected the flight efficiency parameters. Consequently, the average changes in the flight efficiency parameters were also only calculated for the rerouted aircraft and did not include each simulated aircraft of a run. The number of rerouted aircraft in the dynamic simulation is listed in Table 12.

**Table 12.** Number of rerouted aircraft in the dynamic simulation.

| Demand Scenario | Number of Rerouted Aircraft |
|---|---|
| I | 87 |
| II | 313 |
| III | 319 |

A summary of the average changes of the three flight efficiency parameters in the dynamic simulation is presented in Table 13, and these values give a global overview of the impact in the UK–Ireland region. The results show that the average changes in flight distance, flight duration, and fuel consumption had a decrease from demand scenario I to II and an increase from II to III. Interestingly, this effect was noticeably less strong for the changes in fuel consumption.

**Table 13.** Overview of the flight efficiency parameters in the dynamic simulation (average values).

| Demand Scenario | Additional Flight Distance (in NM) | Additional Flight Duration (in min) | Additional Fuel Consumption (in kg) |
|---|---|---|---|
| I | 9.28 | 1.26 | 98.57 |
| II | 7.44 | 0.98 | 98.02 |
| III | 7.96 | 1.05 | 102.71 |

In addition to the average changes, the total changes in the flight efficiency parameters were calculated. The corresponding results are shown in Table 14. As expected, the summed values of the parameters increase through each demand scenario.

**Table 14.** Overview of the flight efficiency parameters in the dynamic simulation (total values).

| Demand Scenario | Additional Flight Distance (in NM) | Additional Flight Duration (in min) | Additional Fuel Consumption (in kg) |
|---|---|---|---|
| I | 807.70 | 109.83 | 8575.47 |
| II | 2328.10 | 306.10 | 30,679.50 |
| III | 2539.60 | 333.58 | 32,763.18 |

Additionally, the changes in the aircraft's flight efficiency parameters because of each single use case was determined, which allows a designation of the impact per use case. Table 15 shows the average changes in the flight efficiency parameters per use case, whereas Table 16 lists the total changes. It can be seen that three use cases ("2_04_UC_LAUN_DE1", "2_05_UC_LAUN_DE1", and "2_10_UC_FORB_RV1") did not have interactions with the conventional air traffic during their scheduled interval, so there was not an impact on the air traffic from these use cases. The two variants of use case "2_10_UC_LAUN_AE1" induced the highest average additional flight distance, flight duration, and fuel consumption, but only two or four aircraft were rerouted around the polyhedrons of this use case. The two HAPS use cases had the largest number of rerouted aircraft, with "2_26_UC_HAPS_AC1" causing the highest total amount of additional flight distance, flight duration, and fuel consumption.

**Table 15.** Overview of the flight efficiency parameters per use case in the dynamic simulation (average values).

| Use Case | Additional Flight Distance (in NM) | Additional Flight Duration (in min) | Additional Fuel Consumption (in kg) | Number of Rerouted Aircraft |
|---|---|---|---|---|
| 2_00_UC_ATOB_HA1 | 9.50 | 1.30 | 53.87 | 1 |
| 2_04_UC_LAUN_DE1 | - | - | - | 0 |
| 2_05_UC_LAUN_DE1 | - | - | - | 0 |
| 2_06_UC_LAUN_DE1 | 7.90 | 0.99 | 32.92 | 4 |
| 2_07_UC_HAPS_AC1 | 8.44 | 1.15 | 93.05 | 82 |
| 2_07_UC_LAUN_AE1 | 0.30 | 0.03 | 3.53 | 1 |
| 2_10_UC_ATOA_AL1 | 4.80 | 0.77 | 40.37 | 3 |
| 2_10_UC_FORB_RV1 | - | - | - | 0 |
| 2_10_UC_LAUN_AE1 (Northern trajectory) | 50.45 | 6.63 | 412.20 | 2 |
| 2_10_UC_LAUN_AE1 (Southern trajectory) | 50.42 | 6.54 | 506.57 | 4 |
| 2_26_UC_HAPS_AC1 | 6.59 | 0.85 | 97.34 | 226 |

**Table 16.** Overview of the flight efficiency parameters per use case in the dynamic simulation (total values).

| Use Case | Additional Flight Distance (in NM) | Additional Flight Duration (in min) | Additional Fuel Consumption (in kg) |
|---|---|---|---|
| 2_00_UC_ATOB_HA1 | 9.50 | 1.30 | 53.87 |
| 2_04_UC_LAUN_DE1 | - | - | - |
| 2_05_UC_LAUN_DE1 | - | - | - |
| 2_06_UC_LAUN_DE1 | 31.60 | 3.97 | 131.69 |
| 2_07_UC_HAPS_AC1 | 692.40 | 94.27 | 7629.97 |
| 2_07_UC_LAUN_AE1 | 0.30 | 0.03 | 3.53 |
| 2_10_UC_ATOA_AL1 | 14.40 | 2.30 | 121.10 |
| 2_10_UC_FORB_RV1 | - | - | - |
| 2_10_UC_LAUN_AE1 (Northern trajectory) | 100.90 | 13.27 | 824.40 |
| 2_10_UC_LAUN_AE1 (Southern trajectory) | 201.70 | 26.15 | 2026.28 |
| 2_26_UC_HAPS_AC1 | 1489.60 | 192.45 | 21,998.38 |

The changes in the flight efficiency parameters can be summarized according to the use cases' main categories, which are presented in Table 17 for the average changes and in Table 18 for the total changes. The most remarkable result to emerge from this summary is that launchers caused the highest impact on average, whereas HAPS operations caused the highest impact regarding the summed changes. This means that launchers caused a high impact for a small number of aircraft (high impact for a single flight), while HAPS affected many aircraft with slight changes regarding their flight efficiency that caused a high impact on ATM level.

**Table 17.** Overview of the flight efficiency parameters per use case category in the dynamic simulation (average values).

| Use Case Category | Additional Flight Distance (in NM) | Additional Flight Duration (in min) | Additional Fuel Consumption (in kg) | Number of Rerouted Aircraft |
|---|---|---|---|---|
| HAPS | 7.08 | 0.93 | 96.20 | 308 |
| LAUN | 30.41 | 3.95 | 271.45 | 11 |
| ATOA | 4.80 | 0.77 | 40.37 | 3 |
| ATOB | 9.50 | 1.30 | 53.87 | 1 |
| FORB | - | - | - | 0 |

**Table 18.** Overview of the flight efficiency parameters per use case category in the dynamic simulation (total values).

| Use Case Category | Additional Flight Distance (in NM) | Additional Flight Duration (in min) | Additional Fuel Consumption (in kg) |
|---|---|---|---|
| HAPS | 2182 | 286.72 | 29,628.35 |
| LAUN | 334.50 | 43.42 | 2985.90 |
| ATOA | 14.40 | 2.30 | 121.10 |
| ATOB | 9.50 | 1.30 | 53.87 |
| FORB | - | - | - |

A 1-tailed paired *t*-test was performed to determine statistical significance regarding the differences in flight efficiency. The comparison was performed between the output parameters of the dynamic simulation runs and the static simulation with a significance level $\alpha$ of 5%. The test revealed that the changes regarding the aircraft's flight efficiency parameters because of HAOs were statistically significant in each dynamic simulation run.

## 7. Discussion

This study outlines an impact assessment of HAOs on the air traffic in Europe. Therefore, a synthesis of applicable HAOs was conducted that served as a basis for a simulation-based analysis of the impact of HAOs on conventional air traffic. Two different types of fast-time simulations were performed: a static simulation and a dynamic simulation. In this section, the results of the impact assessment are discussed and interpreted.

The results need to be treated with care since they strongly rely on the simulations' input, as outlined in Section 5.2. Regarding the static simulation, the values are in line with our expectations, which means that HAPS caused the most interactions with the conventional air traffic. This can be explained by the large dimensions of their polyhedrons. Although launchers had the largest number of use cases, they caused the second most interactions with the air traffic. A reasonable explanation for this effect is the smaller size of the launchers' polyhedrons compared to HAPS. FORB operations showed the fewest interactions because their polyhedrons had the smallest size and only one FORB use case was present in the simulation.

As mentioned in Section 6.2, the results of the dynamic simulation showed that the average changes in the flight efficiency parameters decreased from demand scenario I to II and increased from II to III. This effect can be explained by the number of rerouted aircraft

per demand scenario. From demand scenario I to II, there was a large increase of 226 additional rerouted aircraft that had smaller changes in flight distance and duration than the 87 aircraft in demand scenario I. This led to a decrease in the average. It is interesting to note that this effect was less strongly noticeable regarding the changes in fuel consumption. A look into the aircraft types of the additional rerouted aircraft showed that these were mainly wide-body aircraft (A332, A333, B744, B763, B772, B77W, B788, and B789) from the North Atlantic Tracks that had a high level of fuel consumption. In Table 19, the aircraft types of the affected flights in demand scenario II and III are presented according to their ICAO type designators.

**Table 19.** Aircraft types of additional rerouted flights in demand scenarios II and III.

| Aircraft Type | Quantity |
| --- | --- |
| A310 | 2 |
| A320 | 2 |
| A321 | 1 |
| A332 | 16 |
| A333 | 29 |
| A346 | 4 |
| A359 | 3 |
| A388 | 8 |
| AT76 | 3 |
| B737 | 2 |
| B738 | 7 |
| B744 | 18 |
| B748 | 4 |
| B752 | 5 |
| B763 | 25 |
| B764 | 7 |
| B772 | 22 |
| B77L | 5 |
| B77W | 17 |
| B788 | 14 |
| B789 | 13 |
| B78X | 3 |
| C56X | 4 |
| CL60 | 1 |
| DH8D | 2 |
| E35L | 2 |
| F2TH | 2 |
| FA50 | 1 |
| G280 | 1 |
| GL5T | 1 |
| GLEX | 3 |
| GLF4 | 1 |
| GLF5 | 2 |
| GLF6 | 1 |
| MD11 | 1 |

In order to understand how the simulation results should be evaluated, the data in Tables 15 and 16 were compared to real changes in flight efficiency parameters that were a consequence of airspace closures in the past. After an analysis of NOTAMs [16], two exemplary events were identified that entailed a temporary closure of airspace in Europe. The first event (NOTAM A4079/18) occurred on 2 November 2018 from 7:00 a.m. to 2:00 p.m. UTC and featured a warning about rocket testings west of the Norwegian coast. The impacted area is presented in Figure 9a. The second event (NOTAM A1362/19) occurred on 15 November 2019 from 3:00 a.m. to 12:32 p.m. UTC (early termination) and featured an activation of the restricted area "ESR01" at the Esrange Space Center because of rocket firings. Figure 9b illustrates the dimensions of "ESR01".

Changes in the flight efficiency parameters were identified using matching days where the air traffic was not affected by the airspace closures in Figure 9. The criteria for selection were an identical weekday and the same flight plan period (winter season) to ensure that there were a similar number of flights and a similar amount of recurring scheduled traffic. For NOTAM A4079/18 (2 November 2018), 30 November 2018 was chosen as a reference date, while NOTAM A1362/19 (15 November 2019) had 8 November 2019 as reference date. The historic air traffic data of the respective days were obtained from the DDR2 by EUROCONTROL [21]. Since flight duration, and therefore fuel consumption, is dependent on weather elements such as wind speed, only changes in flight distance were calculated from the real data because the simulation model did not include weather elements. Consequently, only the values of changes in flight distance can be compared. To identify flights that were rerouted on the specific days with airspace closures, an analysis was performed by applying the spatial and temporal dimensions of the areas specified in NOTAMs to their reference date. Afterwards, it was possible to identify flights whose trajectories intersected the areas in the appropriate time window. The length of these trajectories served as an unimpeded baseline. In a next step, the air traffic data of the days with airspace closures were analyzed to find recurring flights that were also scheduled on the reference date. The criteria for recurring flights were an identical callsign and an identical departure and arrival airport. Rotorcraft were excluded. The analysis showed 23 flights whose trajectories intersected the area in the appropriate time window of NOTAM A4079/18 on the reference date 30 November 2018. Of these 23 flights, 20 flights could be found as recurring flights on 2 November 2018 that were rerouted around the impacted area. Regarding NOTAM A1362/19, three flights were identified whose trajectories intersected the area in the appropriate time window on the reference date 8 November 2019. All three flights could be found as recurring flights on 15 November 2019. These trajectories served as impeded trajectories because of reroutings. To measure changes in the flight distance, the length (from departure to arrival airport) of the baseline trajectories was subtracted from the length of the impeded trajectories. Table 20 shows the changes in the flight distances from historical air traffic data because of airspace closures.

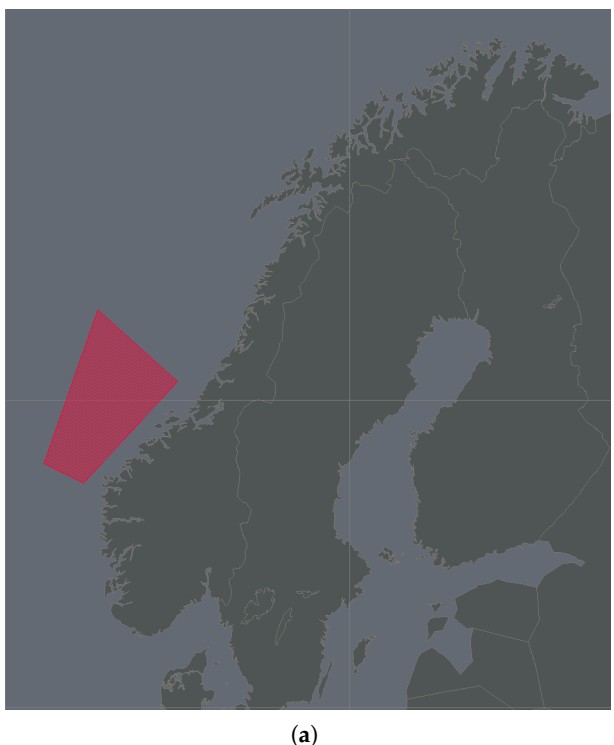

(a)

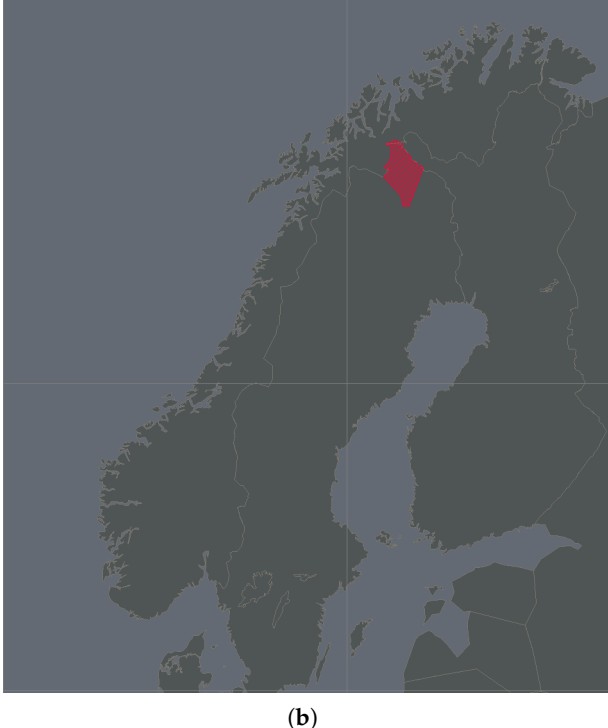

(b)

**Figure 9.** Restricted areas of airspace closures: (**a**) NOTAM A4079/18; (**b**) NOTAM A1362/19.

**Table 20.** Changes in the flight distances from historical air traffic data (average and total values).

| NOTAM | Average Additional Flight Distance (in NM) | Total Additional Flight Distance (in NM) | Number of Rerouted Aircraft |
|---|---|---|---|
| A4079/18 | 107.37 | 2147.50 | 20 |
| A1362/19 | 32.69 | 98.07 | 3 |

The analysis shows that the area of NOTAM A4079/18 lead to much larger deviations and a larger number of rerouted aircraft than the area of NOTAM A1362/19. This substantiates the simulation results, where it was derived that HAOs of smaller dimensions have a lower impact than HAOs of larger dimensions. Additionally, the location has an influence on the impact since NOTAM A1362/19 is located in a more northern territory with a smaller amount of air traffic flying through this region. The changes in the flight distances from historical air traffic data revealed that the additional flight distances tend to be larger on average and in total compared to the simulation results of launchers. The cause of this effect is a consequence of the modeled routing in the simulation, where the aircraft flew along the great circle distance between their starting point and endpoint. In reality, aircraft fly on predefined airways or directly between waypoints, which is always longer than the great circle distance. Another reason is that most of the rerouted flights had a departure or arrival airport in North America and had to join a specific so-called 'North Atlantic Track' [23] on their way over the North Atlantic. As a consequence, the flight distance was additionally extended when the aircraft flew around the impacted areas. The comparison of the simulation results with historical air traffic data revealed that the simulation results can slightly underestimate the real impact on the air traffic under the modeled conditions.

The values in Tables 15–18 underline that the impact of HAOs on the air traffic can be clearly reduced if HAOs operate, whenever possible, during intervals without air traffic, which is also one of the most important conclusions that should be drawn from the impact assessment. Another relevant observation is that launchers had the largest average impact on the conventional air traffic. The reason for this result is the large shape of their corresponding polyhedrons. Due to their function of protecting the air traffic from debris, the shape of these polyhedrons is very long and nearly perpendicular to the air traffic, which led to broad diversions. Nevertheless, the polyhedrons of launchers are located in the north of Europe, which is an area of low air traffic density, and hence, only a few aircraft were affected in total. This is contrary to HAPS, which caused large deviations in the flight efficiency parameters in total because of their location in UK and Ireland, where the so-called 'North Atlantic Tracks' [23] start and end. Therefore, the traffic density in this region is very high, and a large number of aircraft were affected. An additional important reason for the total impact is the long duration of restricted airspaces in the case of HAPS.

In each dynamic simulation run, five aircraft could not be rerouted in each case because their arrival airports were inside a polyhedron of the use case "2_10_UC_ATOA_AL1" or "2_07_UC_HAPS_AC1". A different approach would be needed to solve this interaction, such as delaying the aircraft at their departure airport while still being on ground or making an in-flight speed adjustment.

## 8. Conclusions

This paper has provided an impact analysis focusing on an assessment of the potential impact of specific HAO vehicle types on existing air traffic, considering specifically the interactions with air traffic in airspaces below FL660. For this purpose, a fast-time simulation was conducted. Two types of simulations were performed featuring one day of air traffic, and output parameters regarding flight efficiency were specified.

Afterwards, the changes in these parameters were calculated and compared between the simulation without interference from HAOs (static simulation) and with interference from HAOs (dynamic simulation). Off-nominal events, such as structural or system failure,

were not the focus of this study. The impact of HAOs on the existing air traffic was measured for the nominal operations of HAVs and by means of changes regarding the existing air traffic's flight distance, flight duration, and fuel consumption. Since only changes in the flight efficiency parameters were examined, this research is a kind of preliminary simulation of HAOs in Europe.

The following core statements, which depend on the modeling characteristics and limitations described in Section 5.2, could be determined concerning the impact of HAOs on existing air traffic. These statements summarize the various results of the simulation runs and aim to derive fundamental assertions.

- The impact on the conventional air traffic differs depending on the type of HAOs.
- Launchers cause the highest impact on average per affected flight, whereas HAPS cause the highest impact in total of all affected flights.
- The impact increases with the number of locations of HAOs.
- The location of HAOs influences the impact factor on the air traffic. The further to the north of Europe the HAOs are located, the fewer interactions between HAOs and conventional air traffic occur.
- HAOs at night (between sunset and sunrise) impact a smaller number of flights than HAOs at daytime (between sunrise and sunset).

This study showed that the changes in the flight efficiency parameters were statistically significant for each dynamic simulation run. Regarding the dynamic simulation, it was revealed that there is a difference between the impact for a single flight and the impact at the ATM level. This means that launchers caused a high impact on some flights but a lower overall impact on the air traffic because only a few flights were affected from these operations, whereas HAPS caused a large number of interactions with conventional air traffic, leading to large total changes in flight efficiency. In conclusion, the impact analysis showed that the duration of HAOs had an appreciable influence on the impact because coordinating time windows for HAOs can reduce their impact on the existing air traffic notably.

Concerning the main categories of the vehicle use cases, HAPS operations were most critical to the conventional air traffic in the case of nominal operations because of their large dimensions and long durations/low speeds. Since the HAPS operations were based on a specific vehicle, other results can be determined with different input data. Launchers, A-to-A, A-to-B, and from-orbit operations can take place with a low impact on the existing air traffic by means of coordination between the stakeholders.

The scope of this impact analysis focused on nominal events, so the results are dependent on the assumptions described in Section 5.2. Examining off-nominal events might lead to deviating findings. Furthermore, the impacts were dependent on the polyhedrons defined for the purpose of the simulation. It has to be noted that especially for HAPS, hypersonic A-to-B, and from-orbit operations, it can be assumed that each polyhedron may not fit with the airspace volume required for the related HAO since the mandatory air risk analysis (see Commission Implementing Regulation (EU) 2017/373) related to the operation authorization from the ATM perspective includes additional factors, such as contingency procedures and emergency situations that may extend the volume when considering off-nominal events. It is noteworthy that the time and place of operation over the complex and dense airspace of the European Civil Aviation Conference area should be considered for segmentation.

**Author Contributions:** Conceptualization, O.P., L.L., S.L. and S.K.; methodology, O.P., L.L. and S.K.; software, O.P.; validation, O.P., L.L., S.L. and S.K.; formal analysis, O.P. and S.L.; investigation, O.P. and L.L.; resources, O.P., L.L., S.L. and S.K.; data curation, O.P. and L.L.; writing—original draft preparation, O.P. and L.L.; writing—review and editing, S.L. and S.K.; visualization, O.P. and L.L.; supervision, S.K.; project administration, S.K.; funding acquisition, S.K. All authors have read and agreed to the published version of the manuscript.

**Funding:** This research was funded by the European Union's Horizon 2020 research and innovation program under Grant Agreement No. 890417. The APC was funded by the DLR publication fund.

**Institutional Review Board Statement:** Not applicable.

**Informed Consent Statement:** Not applicable.

**Data Availability Statement:** Data are available from the corresponding author (Oliver Pohling). Access to the Aviation Data for Research Repository (https://www.eurocontrol.int/dashboard/rnd-data-archive (accessed on 30 May 2022)) as well as the Demand Data Repository (https://www.eurocontrol.int/ddr (accessed on 30 May 2022)) has to be granted by EUROCONTROL.

**Conflicts of Interest:** The authors declare no conflict of interest.

## Abbreviations

The following abbreviations are used in this manuscript:

| | |
|---|---|
| AirTOP | Air traffic optimizer |
| AOI | Area of interest |
| ATC | Air traffic control |
| ATM | Air traffic management |
| ATOA | A-to-A sub-orbital flights |
| ATOB | A-to-B sub-orbital flights |
| DDR2 | Demand Data Repository |
| DLR | German Aerospace Center |
| ECHO | European Concept of Higher Airspace Operations |
| EU | European Union |
| FAB | Functional airspace block |
| FABEC | Functional Airspace Block Europe Central |
| FIR | Flight information region |
| FL | Flight level |
| FORB | From-orbit flights |
| HAO | Higher airspace operation |
| HAPS | High-altitude platform system flights |
| HAV | Higher airspace vehicle |
| HS | Hypersonic |
| HTOL | Horizontal takeoff and landing |
| ICAO | International Civil Aviation Organization |
| LAUN | Orbital launchers |
| LTA | Lighter than air |
| NOTAM | Notice to airmen |
| SS | Supersonic |
| TDA | Temporary danger area |
| UAE | United Arab Emirates |
| UK | United Kingdom |
| VTOL | Vertical takeoff and landing |

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
