# Peer review of "Impact of Higher Airspace Operations on Air Traffic in Europe"

_aerospace, doi:10.3390/aerospace10100835_

Round 1
Reviewer 1 Report
References 24 – 31 are not written in the proper way. I suppose that authors have missed it when preparing the paper. Erase that part or adapt the literature in the References and consequently in the text.
The text is very fluent and clear although I have some doubts regarding the methodology since there are many limitations defined in the part 5.2.4. First of all, authors should explain and justify why they use only 1 NM as a rerouting margin from closed airspace since they (row 532 in the text) since they define start/stop rerouting distance on 50 NM. This is in the context that the whole research is based on the possible interactions between conventional flights and closed airspace (which simulate HAO movement).
The second important comment is on simulated rerouted trajectory which implies overall results and calculations of FE indicators. It is not clear how it is done. How did the fast time simulator calculate rerouted trajectories? Which model is used?
Since this research is limited and doesn’t provide any additional results regarding trajectories, air traffic control etc. , it should be stated that it is some kind of preliminary simulation.
In the Conclusion, authors mention EU REG (EU) 2017/373) related to the operation authorisation from the ATM perspective which includes at least additional factors such as contingency procedures and emergency situations that may extend the volume considering off-nominal events. It should be mentioned in the Introduction and added that this research is done only for nominal events with defined assumptions and limitations.
In Abstract authors mention a region in Europe for research while the research is done in 4 different regions.
Commercial space is put as a key word while it is mentioned only twice in the text as well as air traffic management.
Reviewer 2 Report
The author has performed an interesting simulation-based study to assess the impact of Higher Airspace Operations (HAO) on conventional air traffic, using a case study in Europe. This is a timely and meaningful research topic given the increasing interest in utilization of higher airspace. I have the following suggestions to improve the manuscript:
1. Rename Section 2 to "Existing Works."
2. Add a paragraph in the Literature Review summarizing the limitations of previous studies on HAO impacts, and explicitly state this paper's novel contributions.
3. Remove Table 19 since the statistical test results are redundant (the significant differences for each parameter and demand are all “Yes”).
4. Remove or expand on conclusions that seem obvious, such as shorter HAOs having a lower impact. Provide deeper practical implications.
5. Remove references after #24.
6. Fix minor typos and language errors throughout.
Overall this is a clearly written paper that makes a useful contribution to understanding HAO impacts on air traffic. Addressing the above suggestions, especially highlighting the unique contributions, will further improve the manuscript. I recommend acceptance pending these minor revisions and look forward to the revised version. Please feel free to contact me with any questions.

- There are a few typos and minor language errors that need fixing, but overall the manuscript is clearly written.
